# PSEUDO-MASK AND LANGUAGE: A SIMPLE RECIPE FOR OPEN-VOCABULARY SEMANTIC SEGMENTATION

## ABSTRACT

We present a conceptually simple framework for open-vocabulary semantic segmentation, which accurately assigns a semantic label to each pixel in an image from a set of arbitrary open-vocabulary texts. Our method, P-Seg, leverages pseudo-mask and language to train a MaskFormer, and can be easily trained from publicly available image-text datasets. Once trained, P-Seg generalizes well to multiple testing datasets without requiring fine-tuning. Contrary to prior works, P-Seg directly trains for pixel-level feature and language alignment. Without bells and whistles, our method achieves state-of-the-art open-vocabulary semantic segmentation results on three widely tested benchmarks (Pascal VOC, Pascal Context, and COCO). In addition, P-Seg has the extra benefits of scalability with data and consistently improving when augmented with self-training. We believe that our simple yet effective approach will serve as a solid baseline for future research. Our code and demo will be made publicly available soon.

## 1 INTRODUCTION

In recent years, the vision community has made significant strides in improving object detection and image segmentation results, largely thanks to the development of powerful frameworks such as Mask R-CNN (He et al., 2017) and DETR (Carion et al., 2020). These frameworks are known for their intuitiveness, robustness, and flexibility, allowing subcomponents to be easily replaced with better models. In this work, our goal is to develop a similarly enabling framework for open-vocabulary semantic segmentation.

Open-vocabulary semantic segmentation presents a unique challenge as it requires assigning accurate semantic labels to each pixel in an image using arbitrary open-

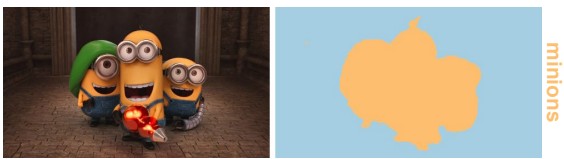

Figure 1: **P-Seg result on a web image.** Our goal is to segment and label every concept, including fictional characters like *minions*.

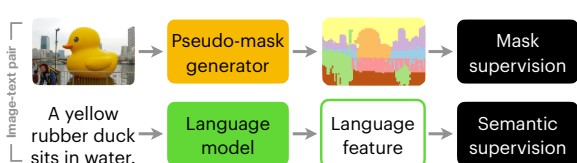

Figure 2: Our **P-Seg** framework leverages pseudo-mask and language to train a MaskFormer. We show that our method of directly training for pixel-level feature and language alignment yields superior results.

vocabulary texts, rather than a fixed set of classes. This means that the model must be able to segment and classify any arbitrary categories expressed in language. To accomplish this, we need both a generalizable grouping model capable of segmenting any object class and a zero-shot classifier capable of classifying objects in an open-vocabulary manner. Moreover, this problem is further complicated by the commonly adopted weakly supervised learning setting where only image-text pairs are used as supervision. Despite these challenges, we show that a surprisingly simple framework can outperform prior state-of-the-art open-vocabulary methods.

Our approach, named P-Seg, is built on top of a MaskFormer model adapted for open-vocabulary segmentation. One of the biggest challenges we face is finding the right supervision since annotated masks and labels are not available. To address this issue, we propose to leverage *pseudo-masks* and *language* to supervise MaskFormer. Our strategy involves using a pseudo-mask generator to provide class-agnostic mask supervision by generating pseudo ground truth masks. We adopt a simple design

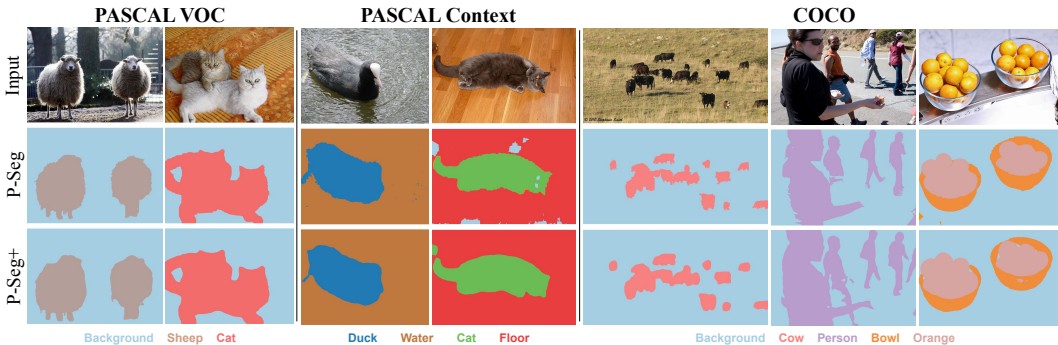

Figure 3: **Qualitative results of P-Seg, evaluated using *all* dataset classes as queries.** Our model copes with challenging situation, such as overlapping objects (col. 2) and small objects (col. 5). Our model is also capable of handling "stuff" categories such as water and floor (col. 3, 4). Moreover, our P-Seg+ model is able to correct small errors observed in the P-Seg method (col. 4). Finally, in the COCO dataset, which featured a significantly higher number of objects, our model is still able to achieve high accuracy in its predictions.

that clusters image representations obtained through self-supervised representation learning methods like DINO (Caron et al., 2021). Our experiments demonstrate that this approach delivers exceptional performance, which is essential for high-quality supervision, as well as rapid processing speed, which is necessary for efficient training. In addition, we use noisy web texts to provide semantic supervision. The image-text dataset contains a wide range of concepts and has demonstrated impressive zero-shot classification results (Radford et al., 2021). We utilize a straightforward image-text contrastive loss, which has proven to be highly effective. Once trained, our model generalizes well to new categories without requiring fine-tuning.

P-Seg is a simple and effective model that can be trained using publicly available image-text datasets, such as Conceptual Captions (Sharma et al., 2018; Changpinyo et al., 2021). This makes it easy to reproduce and extend for further research. Notably, our model does not require any manually annotated segmentation or classification labels for training, nor does it rely on refining existing large image-level alignment models like CLIP (Radford et al., 2021) and ALIGN (Li et al., 2022b). Moreover, P-Seg directly trains a MaskFormer, a well-optimized segmentation model that can predict segmentation maps directly. This is in contrast to many prior works (Zhou et al., 2022; Shin et al., 2022; Luo et al., 2022) that use backbone models optimized primarily for classification (because these algorithms rely on a pretrained image-level alignment model), which can lead to suboptimal results in segmentation. **In this work, we show that directly training for pixel-level feature and language alignment yield superior results.**

The P-Seg framework is also designed to be flexible with easily replaceable submodules. We prioritize *simplicity* in our subcomponent selection to focus on the general design of our framework, while remaining open to more advanced techniques that could result in further improvements. For instance, we can leverage more advanced unsupervised segmentation models such as STEGO (Hamilton et al., 2022) and COMUS (Zadaianchuk et al., 2023) to generate pseudo-masks, while more advanced losses like fine-grained loss (Yao et al., 2021) can provide better semantic supervision. We can also incorporate more advanced segmentation models like Mask2Former (Cheng et al., 2022) to further enhance segmentation performance.

We conducted a thorough evaluation of P-Seg using multiple benchmark datasets, and the results are encouraging. Without bells and whistles, P-Seg surpasses current state-of-the-art open-vocabulary semantic segmentation results on three widely tested benchmarks (Pascal VOC, Pascal Context, and COCO) by a significant margin (an average increase of 4.6% mIoU). In addition, pseudo-mask and language provide scalable supervision and our model consistently improves in performance as more data became available. Finally, we find adding an additional self-training step leads to an even greater improvement to our model, with an average increase of 10.5% mIoU over the previous state-of-the-art method, highlighting the effectiveness of our approach.

## 2 RELATED WORK

**Open-vocabulary segmentation.** The earliest efforts to employ language for image segmentation can be traced back to Duygulu et al.'s seminal work (Duygulu et al., 2002), where the authors tackled

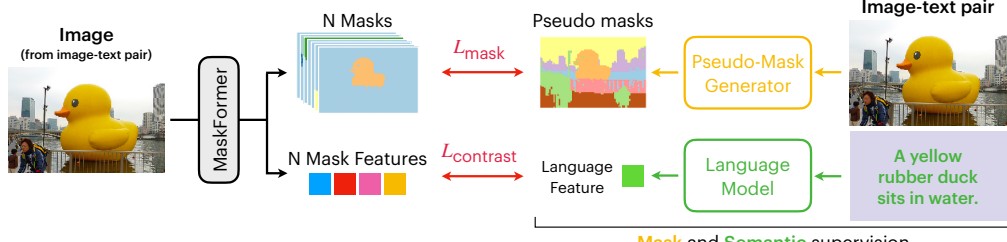

Figure 4: **Overview of P-Seg.** A MaskFormer model computes masks and mask features from an image input. A pseudo-mask generator produces segmentation maps to supervise mask predictions, while a text that describes the image, encoded by a language model trained together with the MaskFormer, provides supervision for mask features using image-text contrastive loss.

image segmentation by framing it as a machine translation problem. The approach employed an EM algorithm to maximize the translation probability $p(t|s)$ that maps image segments to natural language descriptions. Recently, several researchers have explored the use of neural networks for similar segment-word alignment problems (Gupta et al., 2020; Surís et al., 2020; Zhao et al., 2017). Recent advancements in zero-shot approaches (Xian et al., 2019; Li et al., 2020a; Xu et al., 2021; Li et al., 2022a; Ding et al., 2022) aim to develop segmentation models for unseen categories without relying on their pixel-wise labels. However, these approaches still require learning with segmentation labels for a significant number of seen categories, which normally make up 75% to 80% of all categories (see *e.g.* (Li et al., 2022a)). Another line of research uses text as weak supervision and leverages large-scale pre-trained models (Ranasinghe et al., 2022; Ghiasi et al., 2022; Xu et al., 2021; Zhou et al., 2022; Shin et al., 2022) or specialized architectures (Xu et al., 2022; 2023). Our work follows this line of work and only uses text as weak supervision, and differs from prior efforts in three key ways: (1) we directly train a segmentation model, (2) we require no ground truth mask annotation during training, and (3) we train our model without relying on large-scale pre-trained models like CLIP (Radford et al., 2021) or ALIGN (Ghiasi et al., 2022).

**Unsupervised image grouping.** Unsupervised image grouping methods are designed to segment images without the use of manually labeled segmentation masks. Early unsupervised image grouping methods can be roughly categorized as low-level feature-based (Canny, 1986), clustering-based (Kanungo et al., 2002), and graph-based (Shi & Malik, 2000). More recently, self-supervised learning-based approaches (Ji et al., 2019; Cho et al., 2021; Hamilton et al., 2022; Van Gansbeke et al., 2020; 2021; Hwang et al., 2019; Zadaianchuk et al., 2023) have shown superior performance in unsupervised image grouping.

**Vision-language understanding** has generated fruitful research in recent years, largely due to the abundance of image-text paired data. On the one hand, (Lu et al., 2019; Li et al., 2019; Su et al., 2019; Tan & Bansal, 2019; Zhang et al., 2021; Li et al., 2020b; Chen et al., 2020) typically use transformer-based multimodal fusion modules to model the interaction between image and text features and finetune on downstream tasks like VQA (Antol et al., 2015) and NLVR22 (Suhr et al., 2018). They achieve this by using objectives such as masked language/image modeling and image-text matching loss. Alternatively, other works such as (Radford et al., 2021; Jia et al., 2021; Yao et al., 2021; Li et al., 2022b; Mu et al., 2022; Li et al., 2021; Zhai et al., 2022) performs pre-training on large-scale noisy web data using image-text contrastive loss. These models can be directly transferred to image classification tasks with high accuracy.

## 3 APPROACH

Our proposed method, called *P-Seg*, is conceptually simple: we learn a MaskFormer model from *pseudo-mask* and *language*. Our method leverages image-text pairs solely, without relying on ground truth masks or large-scale preatrained models. Figure 4 provides a schematic layout of our approach. In figure 17, we provide pseudocode for the core implementation of training P-Seg.

### 3.1 PROBLEM DEFINITION

We consider the problem of open-vocabulary semantic segmentation, where we aim to learn a function $f$ that maps an image $I$ and a set of category names $C = \{c_i\}$ to a semantic segmentation map $S$, where $c_i$ can be any category name expressed as open vocabulary texts.

Our approach is based on previous works (Xu et al., 2022; Ranasinghe et al., 2022; Zhou et al., 2022), and we adopt their problem setting. Specifically, we use a web dataset of image-text pairs $(I_i, T_i)$ during training, where $T_i$ is a textual label that describes the content of the corresponding image $I_i$. However, since the textual labels are gathered from the web, they may be noisy and contain errors. We do not use any additional manual annotated segmentation or classification labels during training.

During testing, a set of category names $C$ is provided, and the model is tasked with assigning a semantic label $c_i \in C$ to each pixel in an unlabeled image. The performance of the model is evaluated based on its mean Intersection over Union (mIoU) with the ground truth labels.

## 3.2 Adapting MaskFormer

Our approach builds on top of MaskFormer (Cheng et al., 2021). Here, we begin by briefly review MaskFormer and explain the adjustments we made.

The Maskformer model takes an image as input and generates $N$ masks and mask features. First, the input image passes through a backbone model to produce feature maps at different output resolutions. These image features are then fed into a per-pixel encoder, which upsamples and aggregates them into a set of feature maps with higher resolution. Meanwhile, a transformer decoder uses $N$ learnable queries to cross-attend to the set of features with the lowest resolution and gather global information about each segment.

In the original Maskformer, a linear classifier and softmax activation were applied to the output of the decoder to predict class probabilities for a fixed list of categories. However, as we do not have a fixed list of categories, we remove this classifier branch and output the $N$ raw mask features instead.

In addition to predicting mask features, the Maskformer also predicts $N$ binary masks. To predict each mask, a dot product is taken between the mask embedding, generated from mask features, and the high resolution per-pixel feature.

Finally, a combination module takes the raw output, $N$ mask-feature pairs, as input and generates a semantic segmentation map as the output.

## 3.3 P-Seg

P-Seg employs MaskFormer as its segmentation model, but in our weakly-supervised learning setting (where only texts are available), we face the challenge of not having annotated masks and labels. To overcome this, we utilize pseudo labels and language to as supervision.

Our training framework is illustrated in Figure 4. We first generate a set of segmentation maps using our pseudo-mask generator (Sec. 3.3.1) and use them as supervision for mask prediction. Meanwhile, we use a language model to process input text and generate language embeddings. These embeddings provide supervision for mask features by leveraging image-text contrastive loss (Sec. 3.3.2).

Notably, unlike the supervised learning setting, where mask and label annotations are coupled, we *decouple* mask and semantic supervision. This enables us to utilize pseudo-mask and language as two distinct forms of supervision.

In the testing phase (as shown in figure 5), the trained MaskFormer model predicts $N$ masks and mask features from the input image. The language model takes as input a list of candidate category names (represented as texts) and extracts a set of language features. These features are then

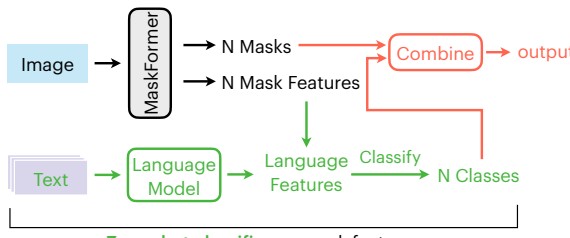

Figure 5: **Testing on P-Seg.** During inference, P-Seg generalize to new categories by leveraging language features generated from a list of candidate classes in text.

used to classify the mask features. This process is similar to the one used in CLIP (Radford et al., 2021), where the image and possible text inputs are encoded by their respective encoders to compute feature embeddings. The cosine similarity between these embeddings is calculated and adjusted by a learnable temperature parameter. The resulting values are normalized into a class probability

distribution using a softmax function, and a combination module is used to takes $N$ mask-class pairs to produce the final segmentation map, similar to (Cheng et al., 2021).

Next, we will provide a detailed description of the subcomponents in our framework.

### 3.3.1 PSEUDO-MASK GENERATOR

In our approach, we use a pseudo-mask generator (fig. 6) to produce a class-agnostic segmentation map from the input image, which supervises the mask prediction of our model.

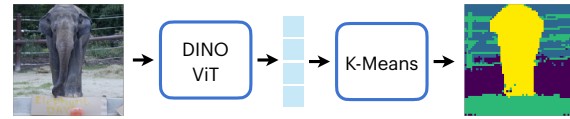

Figure 6: **Pseudo-mask generator** generates pseudo-masks to supervise predicted mask during training. This module takes an image as its input, extracts its features using a DINO pre-trained ViT, and then employs K-means clustering to group the pixels into segments.

To implement the pseudo-mask generator, we adopt a simple strategy that involves clustering tokens extracted from a self-supervised pre-trained ViT. Specifically, we use a DINO-pretrained ViT to compute a set of featurized tokens from the input image. We then apply a clustering algorithm (K-Means in our case) to these tokens, assigning each token a label that corresponds to the index of the cluster it belongs to. We reshape the resulting label map into an image and resize it to the original resolution to supervise the mask prediction of our segmentation model.

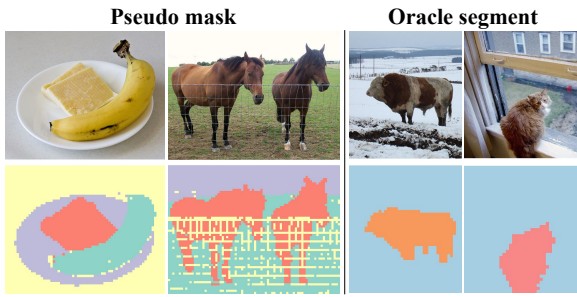

Figure 7: **Example pseudo-masks.** Our pseudo-mask generator is capable of generating high-quality artificial masks. When provided with an oracle label, these masks demonstrate a high degree of overlap with the ground truth annotations.

Despite its simplicity, our pseudo-mask generator achieves both impressive performance, which is crucial for high-quality supervision, and fast processing speed, which is essential for efficient training. We evaluate its performance and compare against baseline methods, and the quantitative results are presented in Table 1, with example predictions visualized in 7. Our method significantly outperforms simple baselines such as K-Means and Spectral Clustering, which naively cluster image pixels, while running two orders of magnitude faster. We also observed that clustering DINO representation outperforms clustering ImageNet pre-trained ViT representation by a significant margin. Notably, our pseudo-mask generator even outperforms GroupViT, which has already employed vision-language training.

Since the predicted masks are unordered, we need to match the $N$ predicted masks with $K$ pseudo ground truth masks. To accomplish this, we utilize bipartite matching, as described in (Carion et al., 2020; Cheng et al., 2021), which assigns a pseudo-mask to each predicted mask such that the overall assignment cost is minimal in all possible assignments. Since each pseudo-mask is assigned to at most one predicted mask, $N - K$

| Method | Sup. | PV↑ | PC↑ | Time(s)↓ |
|---|---|---|---|---|
| Spectral Clus. (Shi & Malik, 2000)* | none | 49.2 | 43.2 | 0.543 |
| K-Means (Kanungo et al., 2002)* | none | 49.5 | 43.3 | 0.188 |
| ImageNet (Dosovitskiy et al., 2020) | label | 68.8 | 58.1 | 0.079 |
| GroupViT (Xu et al., 2022) | text | 73.7 | 54.6 | **0.002** |
| **Ours** | self | **78.8** | **66.3** | **0.002** |

Table 1: **Our pseudo-mask generator achieves excellent oracle performance with rapid speed**, making it an ideal mask supervision. Pascal VOC (PV) and Pascal Context (PC) are evaluated. We report amortised running time on a batch of 128 samples, simulating training time scenario. *We process downsampled image at $\frac{H}{8} \times \frac{W}{8}$ resolution to obtain reasonable running time.

pseudo-masks are unassigned to no-object (∅). Unlike MaskFormer (Cheng et al., 2021), we do not penalize these no-object masks, nor do we use classification loss as an assignment cost. Finally, we compute the mask loss between predicted masks and their corresponding pseudo-mask, utilizing a combination of dice loss (Milletari et al., 2016) and focal loss (Lin et al., 2017).

$$\mathcal{L}_{\text{mask}} = \lambda_{\text{dice}}\mathcal{L}_{\text{dice}} + \lambda_{\text{focal}}\mathcal{L}_{\text{focal}} \tag{1}$$

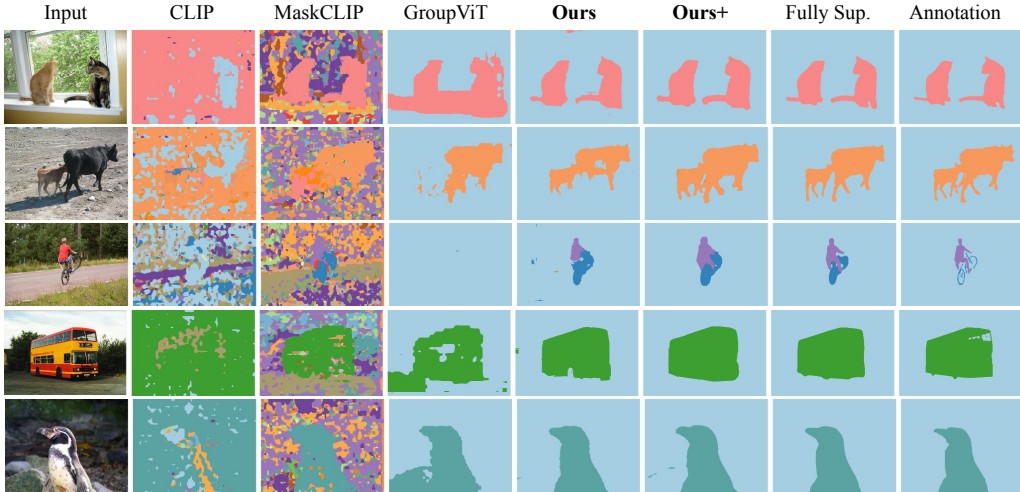

Figure 8: **Qualitative comparison with existing methods.** CLIP (Radford et al., 2021) is primarily designed for classification and does not perform well in segmentation. MaskCLIP (Zhou et al., 2022) adapts CLIP for segmentation, although it produces noisy predictions and cannot handle background classes. GroupViT (Xu et al., 2022) is a strong competitor, but it could struggle in challenging scenarios.

### 3.3.2   LANGUAGE SUPERVISION

Our model learns to classify open-vocabulary concepts from language supervision. To train the model, we use an image-text contrastive loss (Radford et al., 2021; Ghiasi et al., 2022). Specifically, we view $N$ mask features as representation of the input image, each capturing information about a different part of the image. We then compute a single feature that represents the entire image by taking the average of these mask features. To encode the text, we use a text transformer (Vaswani et al., 2017) and select the embedding corresponding to the `[EOS]` token, resulting in a textual feature. Since the visual and textual features may have different dimensions, we project each representation into a common embedding space using 2-layer MLPs. To compute the image-text contrastive loss, we calculate the cosine similarity between the image embeddings and the text embeddings within the same batch. Following common practice (Radford et al., 2021; Mu et al., 2022; Li et al., 2022b), we decouple the image-text contrastive loss into two parts:

$$\mathcal{L}_{I \to T} = -\frac{1}{N} \sum_{i}^{N} \log \frac{\exp(x_i^\mathsf{T} y_i / \sigma)}{\sum_{j=1}^{N} \exp(x_i^\mathsf{T} y_j / \sigma)} \tag{2}$$

$$\mathcal{L}_{T \to I} = -\frac{1}{N} \sum_{i}^{N} \log \frac{\exp(y_i^\mathsf{T} x_i / \sigma)}{\sum_{j=1}^{N} \exp(y_i^\mathsf{T} x_j / \sigma)} \tag{3}$$

where $x_i$ and $y_i$ are L2-normalized embedding of image and text of the i-th pair. $N$ denotes batch size and $\sigma$ is a learnable temperature parameter optimized together with the rest of the model. The total loss is the sum of these two losses, $\mathcal{L}_{\text{contrastive}} = \mathcal{L}_{I \to T} + \mathcal{L}_{T \to I}$. This loss function promotes high similarity for positive pairs and low similarity for negative pairs. The loss is minimized when the positive image-text pairs have the highest similarity. To increase the contrastive efficiency, we aggregate negative samples from all nodes when we use distributed training, enabling more negative samples to be compared against.

### 3.3.3   TRAINING LOSS

Overall, mask loss (Sec. 3.3.1) and image-text contrastive loss (Sec. 3.3.2) complete the necessary mask and semantic supervision that is needed to train our model. The final loss is a weighted combination of the two losses:

$$L = \lambda_{\text{mask}} \mathcal{L}_{\text{mask}} + \lambda_{\text{contrastive}} \mathcal{L}_{\text{contrastive}} \tag{4}$$

In our experiment, we use $\lambda_{\text{mask}} = 1.0$, $\lambda_{\text{contrastive}} = 1.0$, $\lambda_{\text{dice}} = 1.0$, $\lambda_{\text{focal}} = 20.0$.

### 3.3.4 SELF-TRAINING

In order to enhance our results, we introduce an optional step wherein we train a new model using the predictions generated by our current model. This process of self-training results in an augmented model, which we refer to as P-Seg+. More specifically, when we evaluate on a given dataset, we generate pseudo labels for the unlabeled images in the training set. Subsequently, we employ these pseudo labels to train a new segmentation model.

Self-training improves the accuracy by leveraging additional data (Xie et al., 2020), augmentation (Zoph et al., 2020), and bootstrapping (Grill et al., 2020). In our situation, self-training offers even greater benefits since we can take advantage of additional information that is obtainable during testing: unlabeled images and testing categories. We show that this additional step improves our results significantly at no extra manual labelling cost.

## 4 EXPERIMENTS

In this section, we empirically evaluate our method and compare to existing approaches. We show that, although our method is quite simple, it performs surprisingly well against more complex existing methods. We evaluate the open-vocabulary semantic segmentation performance of P-Seg on the validation set of three datasets: Pascal VOC 2012 (Everingham et al., 2009) (21 classes), Pascal Context (Mottaghi et al., 2014) (60 classes) and COCO (Lin et al., 2014) (81 classes). For more implementation details, please refer to our supplementary materials.

### 4.1 SIMPLE BASELINES

The high quality of pseudo-masks (as shown in Figure 6) may lead one to assume that the primary challenge is simply classifying these masks, and that this can be accomplished by utilizing pre-existing methods such as CLIP. To test this assumption, we first develop two simple baselines.

**Baseline 1: Pseudo-mask + CLIP.** Firstly, our pseudo label generator is utilized to obtain pseudo segments. Then, we iterate through all the masks and apply the current mask to the original image. Next, the masked image is fed to CLIP for classification and the resulting class label is assigned to the corresponding segment.

**Baseline 2: Pseudo-mask ViT.** We introduce a new visual backbone that differs from the regular ViT. Instead of pooling all image tokens into a single feature, we first individually pool tokens in each segment of the pseudo-mask into segment features, and then pool these features into a visual embedding. We train a CLIP-like model from scratch using this visual backbone. During testing, we classify each segment feature and assign the label to that segment.

| Method | P. VOC | P. Context | COCO |
|---|---|---|---|
| B1: pseudo-mask+CLIP | 12.9 | 3.9 | 2.9 |
| B2: pseudo-mask ViT | 23.2 | 11.0 | 10.4 |
| **P-Seg (Ours)** | **44.9** | **22.9** | **22.5** |

Table 2: **Simple baselines for open-vocabulary semantic segmentation.** We report results trained on CC12M. All pixels (incl. background) are evaluated. Higher values are better. Two simple baselines fail to obtain satisfactory results, even after using our pseudo masks and no less training data.

The results are presented in Table 2. As we can see, open-vocabulary segmentation is more complex than simply grouping image into segments and then categorizing them into classes, even when the segments are of high quality. Baseline 1 employs a significantly larger pretrained CLIP ViT/L-14 model that was also trained on a much larger dataset, while Baseline 2 is trained using the same data as ours. Nevertheless, both baselines fail to achieve satisfactory results, suggesting that open-vocabulary segmentation cannot be deconstructed in such ways. We hypothesize that a multi-task learning approach that jointly trains the mask and classification tasks could yield significant advantages.

### 4.2 MAIN RESULTS

In table 3, we evaluate our model and compare with existing method on open-vocabulary semantic segmentation task. We observe several key findings: Firstly, our approach outperforms all previous open-vocabulary segmentation methods that does not require mask annotations, as evident from the

| Method | Zero-shot | Supervision | Pascal VOC | Pascal Context | COCO |
|--------|-----------|-------------|------------|----------------|------|
| *Open-vocabulary models (annotated masks required for training):* | | | | | |
| SPNet (Xian et al., 2019) | ✓ | mask+text | 18.3 | 24.3 | - |
| ZS3Net (Bucher et al., 2019) | ✓ | mask+text | 38.3 | 19.4 | 21.1 |
| LSeg (Li et al., 2022a) | ✓ | mask+text | 47.4 | - | 23.4 |
| LSeg (Li et al., 2022a) | ✓ | mask+text | 52.3 | - | 27.2 |
| OpenSeg (Ghiasi et al., 2022) | ✓ | mask+text | 63.8 | 40.1 | - |
| OpenSeg (Ghiasi et al., 2022) | ✓ | mask+text | 77.2 | **45.9** | 38.1 |
| *Open-vocabulary models (annotated masks not required for training):* | | | | | |
| CLIP (Radford et al., 2021) | ✓ | text | 39.6 | 9.0 | 13.8 |
| MaskCLIP (Zhou et al., 2022) | ✓ | text | 49.5 | 25.5 | 23.6 |
| GroupViT (Xu et al., 2022) | ✓ | text | 77.2 | 23.0 | 37.5 |
| **P-Seg (Ours)** | ✓ | text | **81.8** (↑4.6%) | **27.2** (↑4.2%) | **42.4** (↑10.6%) |
| **P-Seg+ (Ours)** | ✓ | text | **84.7** (↑7.5%) | **31.6** (↑8.6%) | **53.0** (↑15.5%) |
| *Fully-supervised segmentation models:* | | | | | |
| DeepLabV3+[†] (Chen et al., 2018) | | mask+label | 89.9 | 48.5 | 66.9 |

Table 3: **Open-vocabulary semantic segmentation results.** [†] denotes our reimplemented results. Higher values are better.

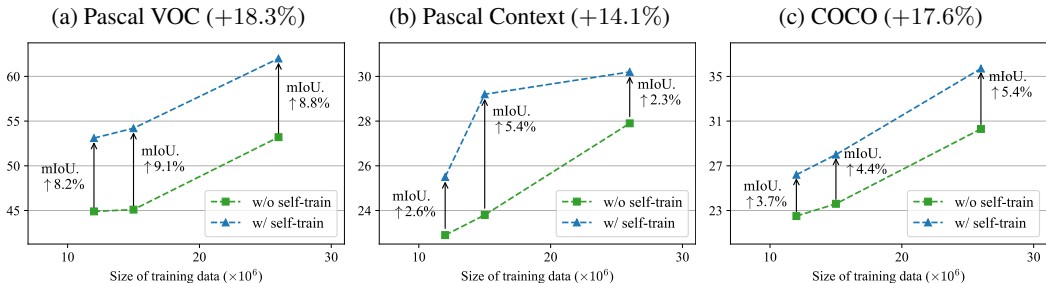

Figure 9: **Self-training improvement.** We show average relative improvement in bracket on top of the plot. we observe that self-training consistently leads to significant improvement for P-seg across all of our training and testing data settings.

table, with a significant margin over the previous state-of-the-art GroupViT (50.5% vs 45.9% mIoU averaged over 3 datasets). Secondly, our self-trained model, P-Seg+, further widens the gap with an impressive 10.5% mIoU improvement over the previous best method (56.4% vs 45.9% 3-avg. mIoU). Finally, Comparing with methods trained with annotated masks, we observe that our method, which learns only from text, comfortably outperforms LSeg. Our model also outperforms OpenSeg on both Pascal VOC and COCO datasets.

### 4.3 EVALUATION WITH BACKGROUND

We also evaluate our model on the evaluation protocol used by (Xu et al., 2022), where the background pixels are included in evaluation. We note that this setting is more difficult because background class is more diverse in appearance and often requires additional processing such as thresholding. Table 4 shows the results. Similar to the previous setting, our P-Seg and P-Seg+ models achieve significantly better performance compared to earlier methods.

### 4.4 ABLATION STUDIES

**Self-training.** We investigated the effectiveness of self-training for improving segmentation performance. To this end, we compared P-Seg and P-Seg+ on three datasets and evaluated the results using Figure 9. We found that self-training consistently improved the segmentation performance by a significant margin (+5.5% mIoU on average), regardless of the data size and test dataset. These results indicate that self-training is a reliable approach for enhancing the performance of P-Seg and can provide a desirable complement for further improvement.

**Data scalability.** To evaluate the scalability of our method, we trained P-Seg and P-Seg+ using three datasets of increasing sizes: 12M, 15M, and 26M. The results of the experiments are presented in Figure 11. We observed that both models achieve significant improvements in performance across all

| Method | Zero-shot | Supervision | Pascal VOC | Pascal Context | COCO |
|---|---|---|---|---|---|
| *Linearly-probed classification models:* | | | | | |
| MoCo v3 (Chen et al., 2021) | ✗ | self | 34.3 | 21.3 | - |
| DINO (Caron et al., 2021) | ✗ | self | 39.1 | 20.4 | - |
| *Open-vocabulary models (annotated masks not required for training):* | | | | | |
| CLIP (Radford et al., 2021)[†] | ✓ | text | 13.5 | 8.1 | 5.9 |
| MaskCLIP (Zhou et al., 2022)[†] | ✓ | text | 26.8 | 22.8 | 12.8 |
| ViL-Seg (Liu et al., 2022) | ✓ | text | 34.4 | 16.3 | 16.4 |
| CLIP$_{py}$ (Ranasinghe et al., 2022) | ✓ | text | 52.2 | - | 25.5 |
| GroupViT (Xu et al., 2022) | ✓ | text | 52.3 | 22.4 | 24.3 |
| GroupViT (Xu et al., 2022) | ✓ | text | 50.8 | 23.7 | 27.5 |
| **P-Seg (Ours)** | ✓ | text | **53.2** (↑2.4%) | **27.9** (↑4.2%) | **30.3** (↑2.8%) |
| **P-Seg+ (Ours)** | ✓ | text | **62.0** (↑11.2%) | **30.2** (↑6.5%) | **35.7** (↑8.2%) |
| *Fully-supervised segmentation models:* | | | | | |
| DeepLabV3+[†] (Chen et al., 2018) | ✗ | mask+label | 78.7 | 46.4 | 55.7 |
| MaskFormer[†] (Cheng et al., 2021) | ✗ | mask+label | 81.2 | 50.0 | 62.1 |

Table 4: **Open-vocabulary semantic segmentation results (incl. background).** [†] denotes our reimplemented results. Higher values are better. When background pixels are included during evaluation, our method also show strong performance.

three testing datasets as the amount of data increased, suggesting that our method scales well with larger datasets.

### 4.5 VISUALIZATION

The qualitative results of our model are illustrated in Figure 3. Our model has demonstrated its ability to handle difficult situations such as overlapping and small objects. Comparing our results to those of existing methods, as shown in Figure 8, we observed that our approach accurately segments objects in challenging cases where previous methods have failed. Additionally, we observed that self-training can correct minor errors in our base model (as shown in detail in fig. 10). In Figure 1 (and 12 in the appendix), we present P-Seg's performance on web images using custom query classes. Our model is able to produce precise results for these categories. For more qualitative results, please refer to our supplementary material.

### 5 CONCLUSION

To summarize, P-Seg is a simple and intuitive framework that enables accurate and generalizable open-vocabulary segmentation. Our algorithm directly trains for pixel-level feature and language alignment, and does not require manual segmentation annotations or extensive pretraining. Once trained, our model outperforms previous open-vocabulary models on three datasets by a substantial margin. Addi-

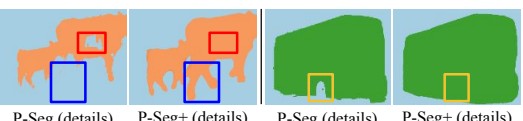

Figure 10: **Visualizing effect of self-training.** Our self-trained P-Seg+ model demonstrates the ability to accurately predict in regions overlooked by P-Seg, as shown in the three colorful rectangles.

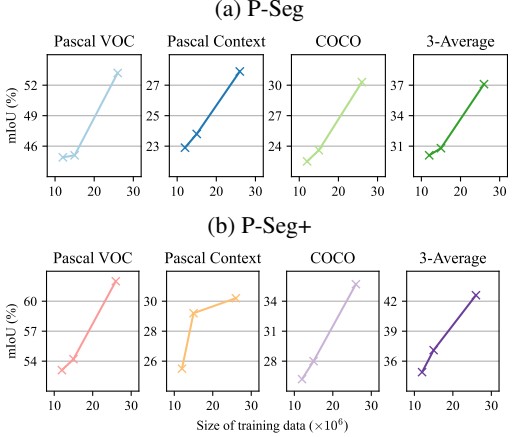

Figure 11: **Scaling training data provide consistent gain in performance, with or without self-training.** We train our model using different sizes of data: CC12M (12M), CC12M+CC3M (15M), and CC12M+CC3M+RedCaps (26M). We note a steady improvement in the model's performance as the data size increases.

tionally, our model demonstrates efficient scalability with increasing data and can be easily augmented by self-training. As computational resources and high-quality vision-language datasets continue to grow, we anticipate that our model will become a highly competitive alternative to close-set methods, providing not only accurate but also flexible segmentation results.

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

# A DISCUSSIONS

## A.1 THE DIFFERENCES BETWEEN P-SEG AND LSEG/OPENSEG/ETC. (PREVIOUS MODELS THAT REQUIRES ANNOTATED MASKS FOR TRAINING)

| Method | Algorithm | Need large VL pretrain | Training data Mask Anno. | Text | Openness data | code | ckpt |
|---|---|---|---|---|---|---|---|
| LSeg OpenSeg | Adapt/Refine image-level VL alignment models | Yes (CLIP) Yes (ALIGN) | Yes (340K) Yes (453K) | 400M 1800M | no CLIP data/code ✗ | ✗ | ✓ ✗ |
| **Ours** | Directly training pixel&language alignment | Not required | Not required | 26M (↓98.5%) | ✓ | ✓ | ✓ |

Table 5: **Differences between P-Seg (Ours) and LSeg (Li et al., 2022a) and OpenSeg (Ghiasi et al., 2022).**

Several key high-level differences are listed in Table 5. Specifically, contrary to LSeg and OpenSeg, which refine image-level models like CLIP/ALIGN, we've found that training for pixel features and language alignment delivers superior results. This is not only a novel method for training open-vocabulary models but also eliminates the need for costly VL pretrainings, streamlining the learning process. Moreover, our model trains without manual mask annotations, thereby reducing supervision needs and offering better generalization. Additionally, we leverage open-source datasets, and will provide full access to all of our source code and pre-trained parameters.

## A.2 THE DIFFERENCES BETWEEN P-SEG AND MASKCLIP/GROUPVIT/ETC. (PREVIOUS MODELS THAT REQUIRES NO ANNOTATED MASKS FOR TRAINING)

| Method | Algorithm | Need large VL pretrain | Backbone | Well-optimized segmentation model | Loss image-level | pixel-level |
|---|---|---|---|---|---|---|
| CLIP MaskCLIP | Adapt/Refine image-level VL alignment models | Yes (CLIP) | ViT | Not used (classification model) | ✓ | ✗ |
| GroupViT | Extract segments from language alignment | Not required | GroupViT | Not used (custom model) | ✓ | ✗ |
| **Ours** | Directly training pixel&language alignment | Not required | MaskFormer | Used | ✓ | ✓ |

Table 6: **Differences between P-Seg (Ours) and CLIP (Radford et al., 2021), MaskCLIP (Zhou et al., 2022), and GroupViT (Xu et al., 2022).**

The main distinctions between our model and previous models that requires no annotated masks for training, such as CLIP, MaskCLIP, and GroupViT, can be found in Table 6. Similar to above, we've discovered that training directly for pixel-level feature and language correlation not only works but actually offers better outcomes. Unlike previous models that rely on classification models (like ViT) or purpose-built custom models (like GroupViT), our approach directly trains a dedicated segmentation model, MaskFormer. This difference often leads previous methods to underperform in segmentation tasks. In order to directly train for pixel feature, we leverage an additional pixel-level mask loss. We show that a simple mask loss obtained from pseudo-masks obtained from raw images is sufficient to yield strong results.

# B ADDITIONAL RESULTS

## B.1 ADDITIONAL DATASETS

We evaluate our method on two new challenging datasets that contain significantly more classes, LVIS (1103 classes) and ImageNet-S (919 classes). The results are shown in Table 7. We observe

| Method | Backbone | 0-shot | Sup. | LVIS (1103 classes) | ImageNet-S (919 classes) |
|---|---|---|---|---|---|
| CLIP (Radford et al., 2021) | ViT-L | ✓ | text | 1.3 | 8.0 |
| MaskCLIP (Zhou et al., 2022) | ViT-L | ✓ | text | 4.3 | 9.1 |
| GroupViT (Xu et al., 2022) | GroupViT | ✓ | text | 7.2 | 32.2 |
| **P-Seg (Ours)** | MaskFormer | ✓ | text | **8.5** | **34.9** |
| Fully Sup. (Dosovitskiy et al., 2020) | ViT-FCN [1] | ✗ | GT | 9.6 | 40.4 |

Table 7: **Open-vocabulary semantic segmentation results on additional datasets.**

(a) **Scaling training data provide consistent gain**: We train our model using different size of data: 12M (CC12M), 15M (+CC3M), and 26M (+RedCaps). We note a steady improvement in the model's performance as the data size increases.

| data | P-Seg | | | P-Seg+ | | |
|---|---|---|---|---|---|---|
| | VOC | Context | COCO | VOC | Context | COCO |
| 12M | 44.9 | 22.9 | 22.5 | 53.1 | 25.5 | 26.2 |
| 15M | 45.1(+0.2) | 23.8(+0.9) | 27.9(+5.4) | 54.2(+1.1) | 29.2(+3.7) | 28.0(+1.8) |
| 26M | **53.2**(+8.3) | **27.9**(+5.0) | **30.3**(+7.8) | **62.0**(+8.9) | **30.2**(+4.7) | **35.7**(+9.5) |

(b) **Self-training offers constant improvement**: We observe that self-training consistently leads to significant improvement on performance across 3 datasets.

| method | 3-Average | | |
|---|---|---|---|
| | 12M | 15M | 26M |
| w/o self-train | 30.1 | 30.8 | 37.1 |
| w/ self-train | **34.9** | **37.1** | **42.6** |
| Δ | +4.8 | +6.3 | +5.5 |

Table 8: **Ablations on data scalability and self-training**. We report mIoU evaluated on three datasets. Higher values are better.

that our model outperforms existing open-vocabulary baseline methods and approaches supervised models, indicating its robustness in challenging scenarios.

## B.2 ABLATION RESULTS

In Table 8, we show numerical results corresponding to Figure 10 and 12 in the main paper. As seen from the table, scaling data and self-training provide consistent gain in performance for our model.

## B.3 PER-CATEGORY RESULT

Table 9 presents the mIoU results of our models and baseline methods on the Pascal VOC dataset, where each class is evaluated separately. Our models outperform current state-of-the-art GroupViT in most classes, and P-Seg+ achieves superior performance across *all* categories. Our models are particularly effective at segmenting large objects such as aeroplanes, buses, and trains, with an average improvement of 11.1 compared to 2.5 for all classes. This suggests that our models benefit from the pseudo-mask generator, which works better for larger objects (showing a 83.3% oracle performance compared to 77.2% for other classes). On the other hand, our self-training model performs better on categories that share consistent texture, such as cats, cows, dogs, and sheep, with an average improvement of 14.3 compared to 8.8 for all classes. This indicates that self-training can identify common features and reduce noise in the self-training labels.

---

[1] We also tried DeepLabV3+ but failed to obtain satisfactory results.

| | | BG. | aeroplane | bicycle | bird | boat | bottle | bus | car | cat | chair | cow | table | dog | horse | motorbike | person | plant | sheep | sofa | train | monitor | mIoU |
|---|---|---|---|---|---|---|---|---|---|---|---|---|---|---|---|---|---|---|---|---|---|---|---|
| OV Methods | CLIP | 13.2 | 10.4 | 4.4 | 8.0 | 5.9 | 19.4 | 27.0 | 17.5 | 26.0 | 3.1 | 19.6 | 9.0 | 21.5 | 16.8 | 11.2 | 11.7 | 5.2 | 13.1 | 7.6 | 21.1 | 12.2 | 13.5 |
| | MaskCLIP | 41.3 | 12.8 | 18.7 | 22.5 | 6.7 | 22.8 | 50.7 | 23.4 | 56.8 | 13.6 | 34.1 | 8.1 | 46.3 | 29.5 | 39.9 | 22.7 | 9.5 | 29.5 | 25.1 | 30.8 | 18.2 | 26.8 |
| | GroupViT | 79.0 | 37.4 | 29.9 | 33.3 | 33.9 | 64.4 | 60.2 | 62.4 | 76.7 | 16.2 | 68.8 | 28.0 | 75.9 | 62.5 | 64.2 | 51.6 | 38.7 | 63.0 | 37.4 | 44.0 | 38.4 | 50.8 |
| | **P-Seg (Ours)** | 81.0 | 47.2 | 40.1 | 38.6 | 30.0 | 63.5 | 74.6 | 67.6 | 75.7 | **18.6** | 65.3 | 34.4 | 72.2 | 56.3 | 68.0 | 50.7 | 45.7 | 60.2 | 33.6 | 53.1 | 41.0 | 53.2 |
| | **P-Seg+ (Ours)** | **86.5** | **53.8** | **42.0** | **48.1** | **49.3** | **76.0** | **84.7** | **74.5** | **87.2** | 17.1 | **81.8** | **35.0** | **83.4** | **65.2** | **74.3** | **65.3** | **46.6** | **78.2** | **40.2** | **58.5** | **53.6** | **62.0** |

Table 9: Per-category open vocabulary semantic segmentation performance over 21 Pascal VOC classes.

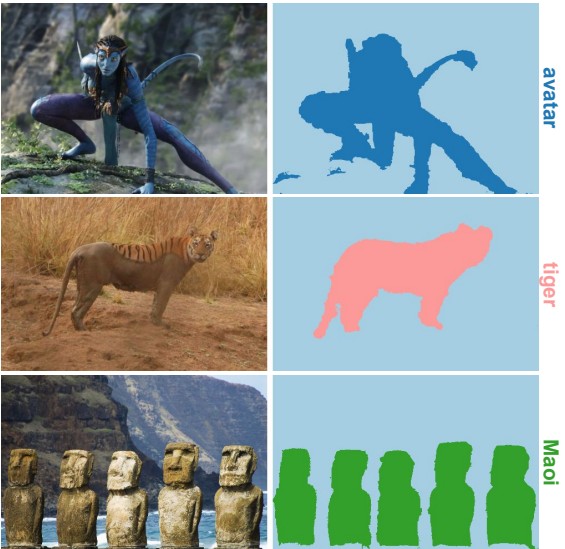

Figure 12: **Qualitative results on web images.** The query class name is shown to the right. Row 1: P-Seg demonstrates its ability to segment fictional characters in an animated scene. Row 2: Despite having taken a mud bath, the tiger can still be easily recognized and segmented. Row 3: P-Seg is capable of identify specific landmarks

## B.4 ADDITIONAL VISUALIZATIONS

Figure 12 presents P-Seg's performance on web images using custom query classes. Figures 13 and 14 present more detailed open-vocabulary segmentation results in higher resolution. As shown in the results, our approach can effectively segment object-centric images from Everingham et al. (2009) (fig. 13) as well as context-rich images from Lin et al. (2014) (fig. 14) accurately. Our method can segment objects based solely on their category name, without requiring any annotations from specific target datasets during training. Figure 15 and 16 provide additional comparison with previous methods.

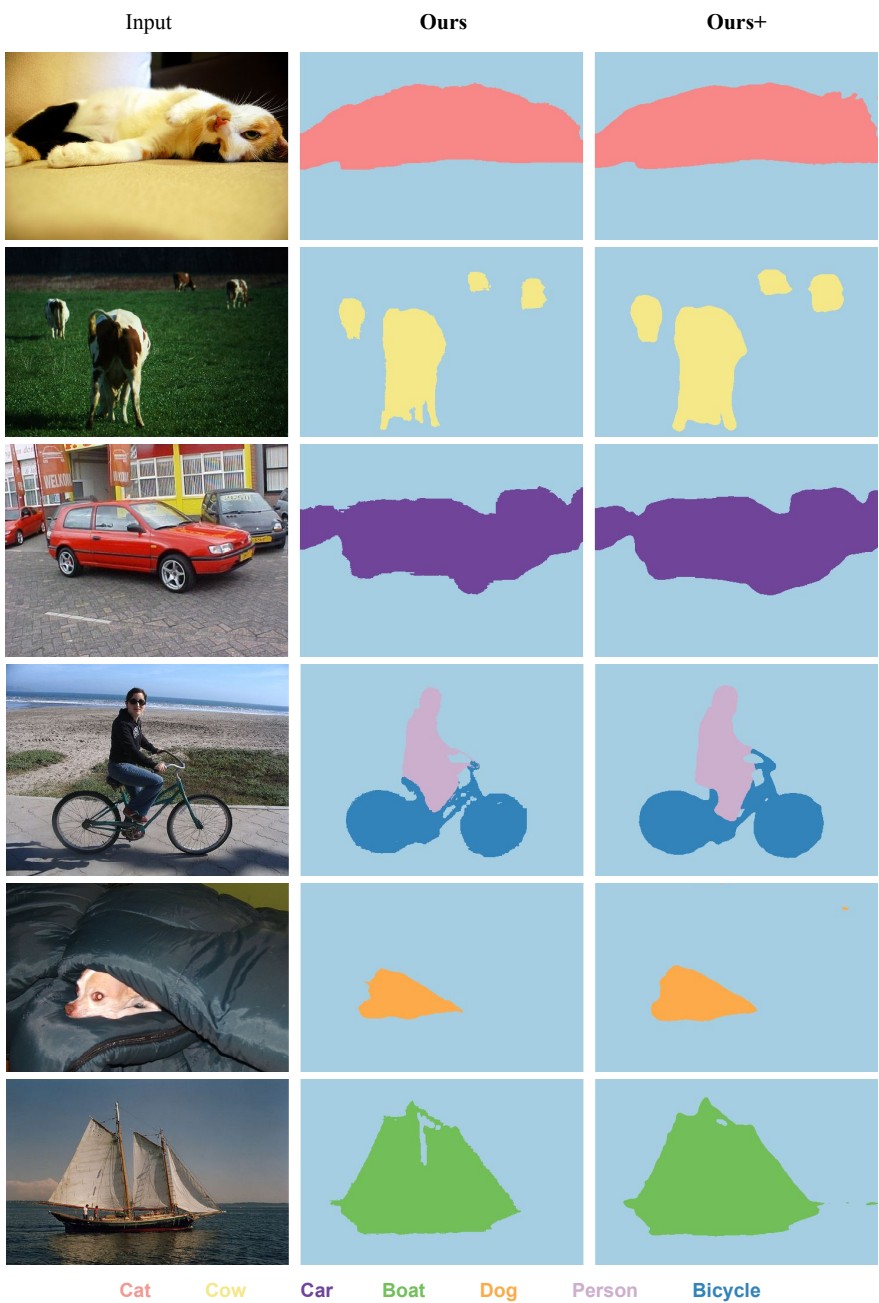

Figure 13: **Additional qualitative results of P-Seg in higher resolution (object-centric images).**
Our method demonstrates robustness in dealing with challenging scenarios, such as objects with
unconventional shapes and poses (row 1), images with unusual color and tone (row 2), objects of the
same class but with differing colors (row 3), objects with the similar color but of different classes
(row 4), concealed objects (row 5), and various other difficult situations.

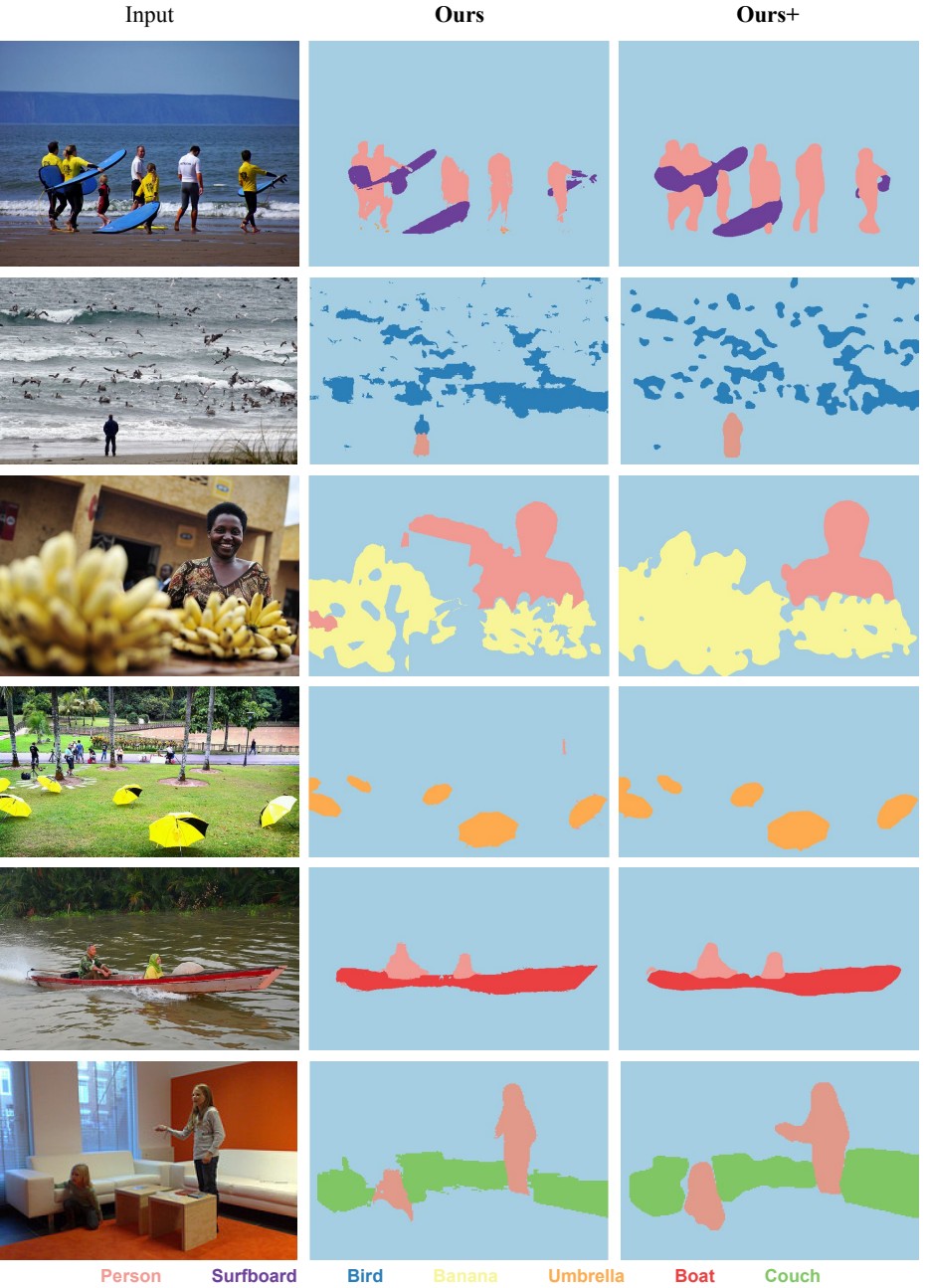

Figure 14: **Additional qualitative results of P-Seg in higher resolution (context-rich images).** Although context-rich images pose challenges in segmentation due to the presence of an increased number of small and cluttered objects, our method can still accurately segment the objects with precision.

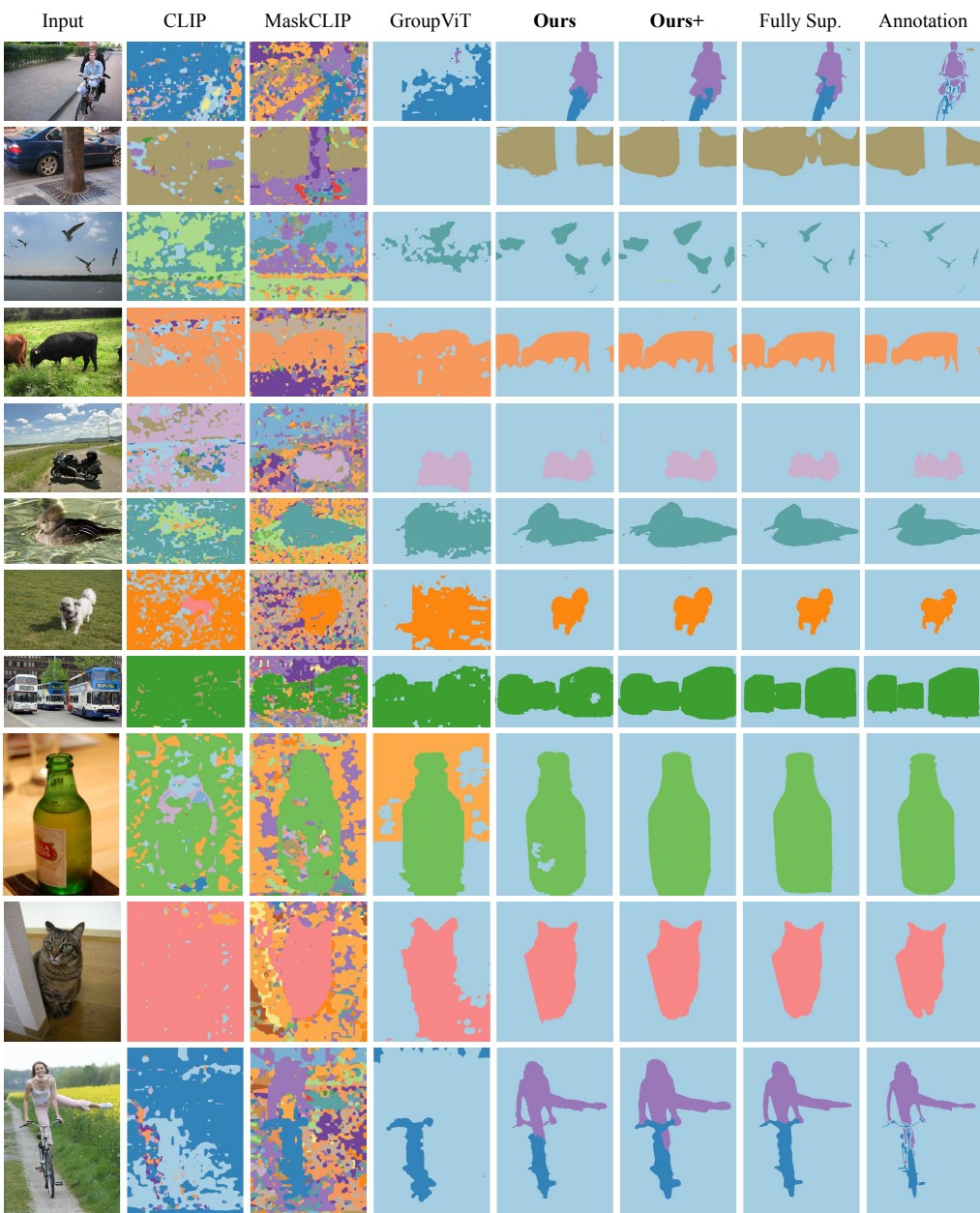

Figure 15: **Additional qualitative comparison with existing methods.** CLIP Radford et al. (2021) is primarily designed for classification and does not perform well in segmentation. MaskCLIP Zhou et al. (2022) adapts CLIP for segmentation, although it produces noisy predictions and cannot handle background classes. GroupViT Xu et al. (2022) is a strong competitor, but it could struggle in challenging scenarios.

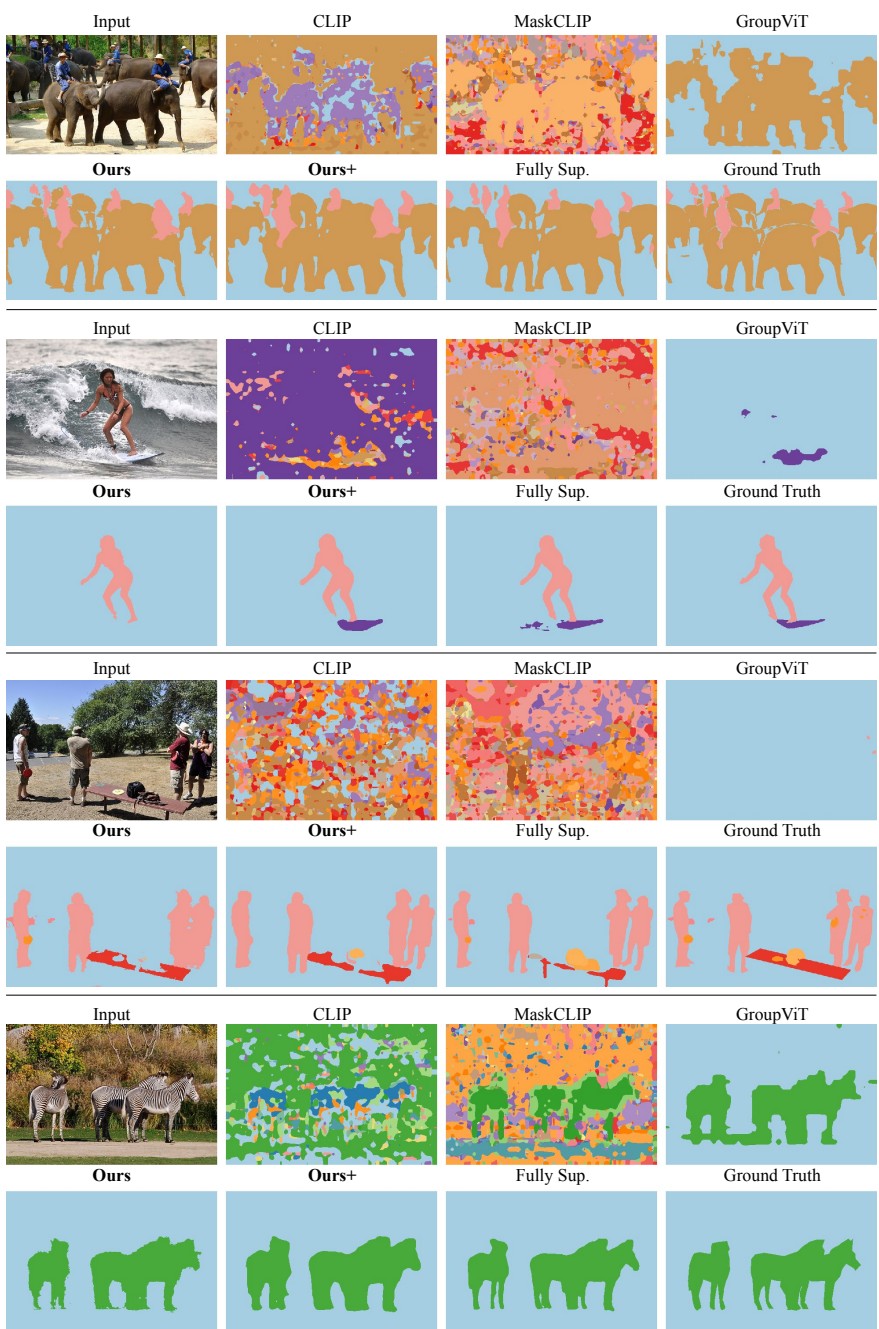

Figure 16: **Additional qualitative comparison with existing methods (continued).**

| config | value |
|---|---|
| optimizer | AdamW Loshchilov & Hutter (2019) |
| base learning rate | 5e-4 |
| weight decay | 0.05 |
| optimizer momentum | $\beta_1, \beta_2 = 0.9, 0.999$ |
| batch size | 4096 |
| learning rate schedule | cosine decay Loshchilov & Hutter (2016) |
| warmup epochs Goyal et al. (2017) | 2 |
| training epochs | 30 |

Table 10: **P-Seg setting.**

| config | value |
|---|---|
| optimizer | AdamW Loshchilov & Hutter (2019) |
| base learning rate | 1e-4 |
| weight decay | 0.05 |
| optimizer momentum | $\beta_1, \beta_2 = 0.9, 0.999$ |
| batch size | 16 |
| learning rate schedule | polynomial decay |
| warmup iters Goyal et al. (2017) | 1.5k |
| training iters | 20k (voc), 40k (ctxt), 80k (coco) |
| layer-wise lr decay Clark et al. (2020) | 0.7 |

Table 11: **P-Seg+ setting.**

## C  IMPLEMENTATION DETAILS

### C.1  P-SEG EXPERIMENTS

**Architecture.** Our experiments use MaskFormer Cheng et al. (2021) with Swin-S Liu et al. (2021) backbone and 6-layer transformer decoder with $N = 64$ queries. The hidden and output feature dimension is 256. The language model is a Transformer Vaswani et al. (2017) with 12 layers, each with a hidden dimension of 256. The context length (maximum length of input text) is set to 77 and the vocabulary size is 49408. We use a 2-layer MLP to project the visual and text feature into a common embedding space of dimension 256. We use DINO ViT-S/8 as the pretrained ViT in pseudo-mask generator which generates $K = 8$ pseudo-masks.

**Training.** During training, we used three publically available datasets: CC3M Sharma et al. (2018), CC12M Changpinyo et al. (2021), and RedCaps Desai et al. (2021), containing 3M, 12M and 12M image-text pairs, respectively. Due to storage constraint, we use only first 11M data samples at a smaller resolution of when using RedCaps dataset. In total, we use at most 26M image-text pairs for training - this is an order of magnitude fewer data than CLIP Radford et al. (2021) and 1-4M fewer than GroupViT Xu et al. (2022). The total dataset takes about 2.4 TB storage space. Table 10 shows our default training setting. All input images are random resized and cropped to $224 \times 224$ in resolution. Following Xu et al. (2022), we extract nouns and verbs from raw sentence because these words are more likely to describe the image.

**Inference.** We evaluate P-Seg on the validation set of three datasets: Pascal VOC 2012 Everingham et al. (2009), Pascal Context Mottaghi et al. (2014) and COCO Lin et al. (2014). The Pascal VOC dataset contain 1449 images for testing. Each image is labeled with 20 foreground classes and a background class. The Pascal Context dataset contains 5104 testing images with 59 foreground classes and a background class. The COCO dataset contains 5000 images for testing with 80 foreground classes and an additional background class. As in Xu et al. (2022), we combine all instances of the same class to get semantic segmentation mask for each image in COCO. Our method of merging individual masks follows GroupViT Xu et al. (2022), except for Pascal VOC, where we employ the original MaskFormer method of semantic inference Cheng et al. (2021). When visualizaing P-Seg, we apply CRF Krähenbühl & Koltun (2011) as an extra post-process step to correct minor errors. During inference, we set the input resolution to $448 \times 448$, which is consistent with GroupViT Xu et al. (2022).

### C.2 P-SEG+ EXPERIMENTS

**Self-training.** For self-training experiments, we use UperNet Xiao et al. (2018) with MAE He et al. (2022) pretrained ViT backbone. We utilize a pyramid-structured network to merge the features obtained from layer 4, 6, 8, and 12 of the ViT, following the implementation of BEiT Bao et al. (2021). We use the same model that we used to evaluate our main results to generate training data from the train set of the respective dataset. Training hyperparameters are provided in Table 11. Following Bao et al. (2021); He et al. (2022), we use a layerwise learning rate decay Clark et al. (2020). We do *not* use relative position embeddings in our backbone ViT model (which is used by Bao et al. (2021); He et al. (2022) at fine-tuning stage).

### C.3 REIMPLEMENTED BASELINES

**CLIP Radford et al. (2021)** We utilized the CLIP ViT-B/16 model along with the official pretraining weights. The ViT model incorporates attentional pooling in its last layer, using an additional [CLS] token to aggregate other tokens. We choose to employ the *value* embedding as the representation of each token, as the query and key embedding of the final layer is not fully trained during CLIP pretraining (only the similarity between the query embedding of the [CLS] token and the key embedding of other tokens is utilized). Finally, we leverage the language model to encode all classes and classify the visual tokens, similar to CLIP's zero-shot classification approach.

**MaskCLIP Cheng et al. (2021)** We use the testing code and weights provided by the authors, but re-evaluating them on the commonly-used protocol that includes the background class. To further assess the efficacy of our approach, as well as baseline methods, we employed the evaluation metric utilized by MaskCLIP, which specifically disregards background pixels.

**GroupViT Xu et al. (2022)** The GroupViT project has provided pre-trained models for two configurations. Without specific clarification, we opt to use the model with the highest average accuracy, which was trained on CC12M, CC15M, and Redcaps datasets. This particular model also closely aligns with our method in terms of training data.

**Fully supervised models (DeepLabV3+ Chen et al. (2018) and MaskFormer Cheng et al. (2021))** We leverage public checkpoints when available. In cases where a checkpoint is not available, we retrain the model using the original training hyperparameters (such as optimizer, learning rate, momentum, and weight decay) along with the standard training schedule, which varies depending on the dataset (40k iterations for Pascal VOC, 80k for Pascal Context, and 160k for COCO). We show the performance of DeepLabV3+ in qualitative comparisons (*Fully Sup.*).

# D PSEUDOCODE

In figure 17, we provide pseudocode for the core implementation of training P-Seg. As shown from the pseudo-code, our method is very simple in concept and has easily replaceable submodules. It contains a concise seven lines of operations. Furthermore, functions such as maskformer, text_encoder, and pmg can be seamlessly updated as more advanced models emerge in the future.

```
# maskformer      - MaskFormer model
# text_encoder    - text transformer
# pmg             - pseudo-mask generator
# I [n, h, w, c] - minibatch of aligned images
# T [n, l]        - minibatch of aligned texts
# N               - number of MaskFormer queries
# C               - number of pseudo masks

# predict mask, mask feature, and text feature
M, M_f = maskformer(I) # [n, N, H, W], [n, N, d_f]
T_f = text_encoder(T)  # [n, d_f]

# aggregate all mask features [n, d_f]
M_f = M_f.mean(axis=1)

# generate pseudo masks [n, C, H, W]
S = pmg(I)

# compute loss
loss_c = contrastive_loss(M_f, T_f)
loss_m = mask_loss(M, S)
loss = (loss_c + loss_m)/2
```

Figure 17: **Pseudocode for training P-Seg with image-text pairs.**

