# OpenReview forum: "Pseudo-Mask and Language: A Simple Recipe for Open-Vocabulary Semantic Segmentation"
_ICLR.cc/2024/Conference — ICLR 2024 Conference Withdrawn Submission_

### Official Review · Reviewer_hdj1 · 2023-10-30

**Soundness:** 1 poor
**Presentation:** 1 poor
**Contribution:** 2 fair
**Rating:** 3
**Confidence:** 4

**Summary:**

The paper presents a method P-Seg and self-training-based extension P-Seg+ for open-vocabulary segmentation. The method trains a MaskFormer segmenter and Transformer language model using a combination of CLIP-like contrastive loss and DETR-style combination of focal and Dice loss with Hungarian matching. The method uses DINO with k-means clustering to obtain pseudo-masks for use with mask losses. Constrative loss uses averaged mask embeddings (produced by a modified MaskFormer class prediction branch) and text embeddings from the text model. The P-Seg model is trained from scratch. At inference time, the mask features are compared to text embedding to assign a class to each mask. The method is evaluated on Pascal VOC, Pascal Context and COCO datasets.

**Strengths:**

1. The presented method achieves competitive performance while training models from scratch (with the exception of using DINO for pseudo-supervision).
2. The paper investigates performance scaling with increasing dataset size, showing improvements.
3. The writing is clear, and the authors promise to release code and checkpoints soon.
4. The method obtains competitive results.

**Weaknesses:**

The central weaknesses of this paper include the limited novelty of the proposal and the need for appropriate positioning and comparison within existing literature, which require further clarification of some claims.

### Novelty
1. The proposed architecture closely follows the proposed scheme of ZegFormer [A] with minor modifications, such as not using a pre-trained language encoder and not using an additional image encoder during inference. The proposed method is not compared to ZegFormer, and the differences are not be discussed.
2. The scheme of pseudo-mask labelling self-training is similarly widely employed; most recently, CutLER [B] proposes to train an unsupervised segmenter using masks obtained from DINO clustering.
3. The use of pseudo-mask generation in the context of OV segmentation has also been explored in DiffuMask [C], which might require further discussion.

### Comparisons and Claims
4. The paper claims the proposed method P-Seg achieves SOTA results, but the comparative baselines could be more recent. Critically, methods such as OVSegmentor [D], TCL [E] and SegCLIP [F] are not included in comparisons. The lack of such baselines is critical because, based on reported results in the paper, P-Seg would not be SOTA on Pascal VOC (OVSegmentor, TCL) and Context (TCL) and COCO (TCL), which contradicts the bolded claim in the introduction. Additionally, OVSegmentor is trained on smaller datasets and similarly does not use CLIP or ALIGN models, relying instead on DINO (like P-Seg), which offers a closely matched baseline.
5. Additionally, the paper compares on the COCO dataset with 80 classes and backgrounds, which matches other papers' descriptions of the COCO-Object dataset. The dataset description might need some additional clarification. CLIPpy results are reported much lower and match those of full COCO (133 classes) rather than COCO-Object (80 classes) from the paper, which might need correcting.
6. It might also be more reasonable to include more recent _mask+text_ methods such as OVSeg [G] or ODISE [H] in the comparison to provide a more complete picture of the models and performance attained in the field.
6. It should also be clarified in what sense are the methods "zero-shot"? Both P-Seg and P-Seg+ likely observe instance images of the target class during training. This is because the word token embeddings of the corresponding classes in the language model were trained, so training datasets should have contained target classes. P-Seg+ is trained on the training sets of the evaluation datasets, which include classes in question by design. Perhaps the paper uses more of a CLIP-like interpretation of "zero-shot", where the model is not trained on the target classes but on a large dataset of images and text that likely _contained_ such instances. This should be clarified. In any case, P-Seg+ should not be classed as zero-shot since it is trained on the train spits of evaluation datasets. As an alternative, observed/unseen class comparisons could be made like in other zero-shot semantic segmentation papers (e.g. [C]).

### Experimental Evaluation
7. The paper lacks ablations that confirm the design choices of the method. The ablation presented centres around showing that the method improves with a larger amount of data. However, this confirms the general trend of deep learning methods and is not, arguably,  very surprising. The question of what makes the method work needs to be addressed, as they are critical to understanding and building on top of this work.
    - Why use simple K-Means on top of DINO features? Prior methods for extracting masks using DINO features have been developed, such as LOST [I], TokenCut [J], and MaskCut [B], which have achieved state-of-the-art results in unsupervised segmentation with mere modification of the clustering procedure. These also offer better points of comparison to the pseudo-mask extraction method.
    - Is setting K=8 critical? This needs to be discussed. How sensitive is the method to this choice? Nouns are extracted from the caption. Why not use the full caption? Why not set K to some function of the number of nouns in the caption?
    - Why are language features compared to predicted mask features? An alternative would be to compare the average features used to obtain dense masks. Is this choice critical?
    - Why is P-Seg+ using a different architecture and different, now pre-trained, backbone? This avoids the opportunity to initialize from large-scale training of the P-Seg. Is the large-scale VL pre-training detrimental in this case?

### Presentation and Clarity
8. The paper also needs some crucial details.
    - The evaluation protocol for including background could be clearer. How is background class determined in P-Seg/P-Seg+? Is textual prompt used for background? What are textual prompts used for the categories? Prior works rely on templates or prompt expansion to obtain textual prompts for classes. Such details should be included.
    - The procedure for self-training should be discussed in detail. In particular, from the description in Sec. 3.3.4, the new model is trained using predictions of the current model, which are described as pixel-dense masks and class labels (Section 3.1, last paragraph). **This makes P-Seg+ a standard closed-set segmentation method, which is not entirely comparable to other methods used in the paper.** If this is the case, then it should be clarified.
9. Presentation of qualitative results should clearly indicate that **P-Seg results are obtained using a CRF**, as it is only mentioned in Appendix C and not when such comparisons are discussed. A comparison could be made without CRF to show substantial differences, as other works might similarly benefit from CRF, and CRF might obscure key differences in results.
10. It also needs to be clarified why specific models in Tables 3 and 4 have multiple entries with different results for the same dataset. How do these models differ? Are these different runs of the same model? This should be clarified.

---
#### References
- [A] "Decoupling Zero-Shot Semantic Segmentation" Ding et al. CVPR 2022
- [B] "Cut and learn for unsupervised object detection and instance segmentation" Wang et al. CVPR 2023
- [C] "DiffuMask: Synthesizing Images with Pixel-level Annotations for Semantic Segmentation Using Diffusion Models" Wu et al. ICCV 2023
- [D] "Learning Open-Vocabulary Semantic Segmentation Models From Natural Language Supervision" Xu et al. CVPR 2023
- [E] "Learning to Generate Text-grounded Mask for Open-world Semantic Segmentation from Only Image-Text Pairs" Cha et al. CVPR 2023
- [F] "Segclip: Patch aggregation with learnable centers for open-vocabulary semantic segmentation" Lou et al. ICML 2023
- [G] "Open-Vocabulary Semantic Segmentation with Mask-adapted CLIP" Liang et al. CVPR 2023
- [H] "Open-Vocabulary Panoptic Segmentation with Text-to-Image Diffusion Models" Xu et al. CVPR 2023
- [I] "Localizing Objects with Self-Supervised Transformers and no Labels" Simeoni et al. BMVC 2021
- [J] "Tokencut: Segmenting objects in images and videos with self-supervised transformer and normalized cut" Wang et al. CVPR 2022

**Questions:**

The key questions have been presented alongside the weaknesses. The following are additional questions that could be considered, though they are very minor.

- Why were the models not pre-trained? Using a pre-trained backbone for a segmentation network is a common practice. It is also not clear why the language model needs training from scratch. It seems that using a language model pre-trained on a large corpus would be beneficial.
- The emphasis seems to be placed on processing time for pseudo-mask generation. However, the method is trained using 30 epochs, which would likely take a lot longer than any pre-calculation of pseudo-masks (single epoch). It is not clear why processing time is so critical.
- The text says that N binary masks are predicted. How are these masks converted to C-way dense masks? How are possible overlaps handled?

---

### Official Review · Reviewer_dUwF · 2023-11-01

**Soundness:** 3 good
**Presentation:** 3 good
**Contribution:** 2 fair
**Rating:** 3
**Confidence:** 5

**Summary:**

The paper presents P-Seg, a simple framework for open-vocabulary semantic segmentation. P-Seg uses pseudomask and language to train a MaskFormer for alignment between pixel-level features and language embeddings. After training, P-Seg generalizes well to multiple testing datasets without fine-tuning. Experiments show that P-Seg achieves encouraging results on severak benchmark datasets (Pascal VOC, Pascal Context, and COCO). Authors also show the scalibility of P-Seg with data.

**Strengths:**

+ The proposed method P-Seg is simple and straightforward which effectively use alignments between pixel-level features and language embeddings as training signals.

+ P-Seg achieves better performance than some baseline methods as well as some open-vocabulary semantic segmentation methods, e.g. LSeg, OpenSeg, MaskCLIP etc.

**Weaknesses:**

- The training objective of P-Seg is similiar to a image-text alignment model, such as CLIP. It is not clear why authors chose to use the text transformer from (Vaswani et al., 2017), rather than the text encoder of CLIP as it may carry similiar propurity from image-text alignment pretraining. Even for pure language model, why not using more recent language models such as Bert or T5.

- Many recent open-vocabulary semantic segmentation works are missing from the referernces, related work and methods discussions, such as [1-4] ,etc. The technical contributions of the proposed method is limited given these recent related studies.

[1] Global Knowledge Calibration for Fast Open-Vocabulary Segmentation, ICCV 2023

[2] Open-Vocabulary Semantic Segmentation with Decoupled One-Pass Networ, ICCV 2023

[3] Open-Vocabulary Semantic Segmentation with Mask-adapted CLIP, CVPR 2023

[4] Side Adapter Network for Open-Vocabulary Semantic Segmentation, CVPR 2023

- Authors claimed state-of-the-art performance for open-vocabulary semantic segmentation. However the comparisons with more recent works such as [1- 4] above are missing. Actually the proposed P-Seg has a large performance gap when comparing with these works. For example, on Pascal Context, P-Seg achieves 31.6 mIoU while [1] is 45.2, [2] is 48.8, [3] is 55.7. On Pascal VOC, P-Seg achieves 84.7 while [1] is 83.2, [2] 91.7, [3] 94.5.

- Results on many important evaluation benchmarks are missing as well, such as ADE20K-150, ADE20K-847, PC-459. It is hard to justify the model's performance on testing cases with more categories.

**Questions:**

Please refer to details in the Weaknesses section above.

---

### Official Review · Reviewer_ddmt · 2023-11-07

**Soundness:** 3 good
**Presentation:** 3 good
**Contribution:** 2 fair
**Rating:** 5
**Confidence:** 3

**Summary:**

- This work presents a simple framework for open-vocabulary semantic segmentation called P-Seg. The proposed method directly trains for pixel-level features and language alignment based on MaskFormer.

 - The simple yet effective framework shows state-of-the-art open-vocabulary semantic segmentation results on several benchmarks, as well as good data scalability and adaptability with self-training.

**Strengths:**

- This paper is well-presented and easy to follow, with the methodology being clear and well-organized. The motivation for using pseudo mask is also clear.

- This paper presents solid quantitative results, as well as a number of qualitative comparisons that clearly show the performance advantage of P-seg. The thorough evaluation also provides detailed investigations into other nice properties of P-seg, such as data scalability.

**Weaknesses:**

- The technical contribution of this work is limited. The authors highlighted the significance of pseudo masks from the DINO pre-trained model. While ablative experiments show its advantage, similar findings on DINO's effectiveness for segmentation have been found in previous works such as [1], which might make the contribution less significant.

 - Another major design of this work starts from the MaskFormer architecture. However, there lack of detailed analysis of this design (See Question).

[1] Self-supervised learning of object parts for semantic segmentation. CVPR 2022

**Questions:**

- What's the design-wise and performance-wise difference between the previous ViT-arch and the MaskFormer-arch in this work? A detailed analysis with other variables fixed is expected.